# Urine-Based Antigen (Protein) Detection Test for the Diagnosis of Visceral Leishmaniasis

**DOI:** 10.3390/microorganisms8111676

**Published:** 2020-10-28

**Authors:** Antonio Campos-Neto, Claudia Abeijon

**Affiliations:** 1DetectoGen Inc., Westborough, MA 01581, USA; cabeijon@detectogen.com; 2Cummings School of Veterinary Medicine, Tufts University, North Grafton, MA 01536, USA

**Keywords:** visceral leishmaniasis, kala-azar, *Leishmania donovani*, *Leishmania infantum*, antigenuria, antigen detection test, monoclonal antibody, diagnostic test

## Abstract

This review describes and appraises a novel protein-based antigen detection test for visceral leishmaniasis (VL). The test detects in patient’s urine six proteins from *Leishmania infantum (chagasi)* and *Leishmania donovani*, the etiological agents of VL. The gold standard test for VL is microscopic observation of the parasites in aspirates from spleen, liver, or bone marrow (and lymph node for dogs). Culture of the parasites or detection of their DNA in these aspirates are also commonly used. Serological tests are available but they cannot distinguish patients with active VL from either healthy subjects exposed to the parasites or from subjects who had a successful VL treatment. An antigen detection test based on the agglutination of anti-leishmania carbohydrates antibody coated latex beads has been described. However, the results obtained with this carbohydrate-based test have been conflicting. Using mass spectrometry, we discovered six *L. infantum*/*L. donovani* proteins excreted in the urine of VL patients and used them as markers for the development of a robust mAb-based antigen (protein) detection test. The test is assembled in a multiplexed format to simultaneously detect all six markers. Its initial clinical validation showed a sensitivity of 93% and specificity of 100% for VL diagnosis.

## 1. Introduction

Visceral leishmaniasis (VL) or kala-azar is a systemic parasitic disease that is endemic in 75 countries with more than 200 million people at risk of infection. It affects 350,000 people each year and kills 20,000–40,000 of them, of which 70% are children under 15 years of age [1].

VL encompasses two different diseases occurring in two historically defined geographical regions of the world: (1) the New World VL (primary South America). In this region the disease is caused by *L. infantum*; and (2) the Old World VL (primarily India Subcontinent). In this region the disease is caused by *L. donovani*. However, it is now known that *L. infantum*, in addition to South America, is the agent of VL that occur in the Mediterranean, Middle East and Central Asia regions of the World. Similarly, *L. donovani*, in addition to the India Subcontinent is the agent of VL in East Africa countries [2]. Notwithstanding the geographical denomination of VL being incorrect, the concepts of New and Old World VL are still commonly and loosely used.

There are no preventive vaccines for human VL. In many areas, VL morbidity and mortality are increasing dramatically due to co-infection with human immunodeficiency virus (HIV) [3,4]. In Southern Europe between 25–75% of all cases of VL emerge from people infected with HIV. Moreover, 1.5–9.5% of all patients with Acquired Immunodeficiency Syndrome (AIDS) suffer from newly acquired or reactivated VL [1,5] in different countries of Europe.

Domestic dogs are a major vertebrate reservoir of *L. infantum* [6], which increases the risk of VL transmission to humans. Canine VL (CVL) is widely present in Latin America (particularly in Brazil) and Southern Europe [7,8], and has been recognized as a potential problem in the US [9,10,11].

A major impediment to achieve an effective management of VL is the lack of an accurate diagnostic test that can distinguish between active VL from asymptomatic parasite exposure [12]. Definitive diagnosis of active VL relies primarily on direct observation of Leishmania parasites in smears or cultures from either liver, or spleen or bone marrow aspirates, which require invasive and risky sampling procedures. In addition, the sensitivity of these tests is in general modest and varies enormously [13,14,15,16,17,18,19]. Nucleic acid amplification tests are also existing tools for the diagnosis of VL. Although these tests are more sensitive than microscopy/culture they are relatively complicated and expensive, and are restricted to referral hospitals and advanced research centers [12,20,21,22]. The serological diagnostic tests currently available measure antibodies against parasitic antigens, and for this reason do not distinguish active disease from simple exposure to the parasite as well as from treated and cured subjects. Serological tests are not suitable for diagnosing active VL for several reasons, which include: (a) high serum antibody levels are present in both asymptomatic and active VL [23,24,25,26,27,28]; (b) serum anti-Leishmania antibodies remain in the circulation for several years after cure, which complicates the diagnosis of relapsed VL [29,30,31]; (c) some individuals from endemic areas with no history of VL have anti-leishmanial antibodies, which is a major hindrance for these tests sensitivity/specificity [32]; and (d) sensitivity of serological tests in VL/HIV co-infected patients is poor, particularly if VL occurs after HIV infection [33,34].

Interesting alternatives to these diagnostic procedures are platforms that detect pathogen antigens (proteins or carbohydrates) in bodily fluids. These platforms also known as antigen detection tests have been successfully used for many years for the diagnosis of several infectious diseases including hepatitis B [35,36] sore throat caused by *Streptococcus pyogenes* [37], pneumonia caused by either *Streptococcus pneumoniae* [38,39] or *Legionella pneumophilla* [40], tuberculosis [41,42], malaria [43,44,45], amoebiasis [46], and COVID-19 [47,48,49].

An antigen detection test for VL was developed approximately 20 years ago [50]. This assay is a latex agglutination test (KATex), which is based on the detection of leishmanial carbohydrate complex antigens using latex beads adsorbed with specific polyclonal rabbit anti-carbohydrate antibody [51]. KAtex has been used intermittently for VL diagnosis as well as a tool for monitoring disappearance of leishmanial antigens after successful treatment of VL [52,53,54,55,56,57]. Recent studies involving VL patients co-infected with HIV have shown that the sensitivity of KAtex varied from 47.7% to 85.7% and specificity from 96–98.7% [58]. These conflicting results can be explained on the grounds of the uncontrolled specificity and sensitivity (affinity/avidity) of the heterogeneous anti-carbohydrate antibodies used in the test [51]. Although KAtex has not been universally used or accepted as a reliable tool for the diagnosis of VL, the results of these clinical studies confirm the feasibility of a urine-based antigen detection test for the diagnosis of VL.

Two protein antigen detection tests have been reported for the diagnosis of VL [59]. Both methods utilize preparations of whole *L. donovani* promastigotes to generate the anti-whole parasite promastigote polyclonal antibodies employed to assemble the tests. However, the sensitivity and specificity of these tests are not adequate for clinical diagnose of VL. In addition, and like KATex, translation of these tests into an actual product is complicated due to the complex nature of the reagents.

In summary, these diagnostic hindrances have serious repercussions for the control of VL for the following reasons: VL is usually fatal if not diagnosed and treated promptly; the effective drugs are toxic, expensive and difficult to administer; and untreated people with VL are reservoirs of infection that put others in their communities at risk. With resistance to first line drugs (antimony) increasing to up to 60% in certain parts of the world the mortality rate from VL is extremely high [60]. Therefore, the development of a simple, reproducible, and accurate diagnostic test of VL should help to overcome or prevent the surge of these vital and complicated outcomes.

To overcome these hurdles, we used mass spectrometry to identified six *L. infantum*/*L. donovani* protein antigens or biomarkers present in urine of humans with active VL (Table 1). These defined markers were then used for the development of an accurate and sensitive protein-based and monoclonal antibody-based antigen detection assay for VL [61,62,63,64,65]. Here, we review in detail the strategy used to discovery these leishmanial biomarkers, the development of a sensitive and specific multiplexed capture ELISA, and the preliminary results of a clinical validation of this new and powerful monoclonal antibody-based antigen detection test for the diagnosis of VL. A recent publication describes six other new proteins of *L. donovani* identified in urine of VL patients using mass spectrometry coupled with immunoprecipitation of parasite antigens [66]. These proteins are not related to the markers described and reviewed herein and their utility as diagnostic markers have not yet been reported.

## 2. Overall Rationale

Targeting pathogen protein antigens that are produced in vivo during disease and eliminated through the urine is a proven idea and strategy in diagnostic development for a variety of infectious diseases including VL [42,67,68,69,70,71]. In addition, the use of urine as a test sample supports the hypothesis that excreted pathogen antigens are produced abundantly in the infected organs during disease and then travel through the blood stream to be eliminated by the urinary system. This means that these antigens are highly stable, which makes them attractive candidates for a diagnostic assay. Therefore, a specific and sensitive antigen detection assay for VL caused by either *L. infantum* or *L. donovani* represents an important innovation not only for the diagnosis of the disease but also for monitoring treatment effectiveness and quickly identifying treatment failures so that alternative therapy can be initiated.

The use of urine as a good alternative specimen for the development of diagnostic test for infectious diseases is summarized in Box 1.

Box 1Advantages of urine as specimen for antigen detection test development for infectious diseases.
Urine contains much less host proteins than plasma or serum, therefore small amounts of molecules from foreign organisms present in this excretion are much more likely to be identified by mass spectrometry than they would in blood.Urine is one of the easiest and least invasive samples that can be collected from a patient. This is particularly important for VL and for example tuberculosis (TB) in children. Gold standard diagnosis of VL requires invasive biopsy of organs like liver, spleen or bone marrow. In TB, because children in general do not spit sputum, invasive gastric lavage is the sample that is used to investigate the presence of *Mycobacterium tuberculosis* in this group of patients.Antigenemia and subsequent antigenuria occurs as a natural consequence of most infectious processes.The presence of pathogen’s antigens excreted in the urine may be detectable sooner than the detection of the pathogen itself in target organs of the disease.Antigen detection in urine is an attractive possibility to facilitate the diagnosis of VL in HIV-co-infected patients because these patients have in general reduced antibody response to *L. infantum/L. donovani* antigens.Antigens found in urine, theoretically, are likely to be highly stable because they have to resist the proteolytic enzymes present in both, target organs of infection and serum. Therefore, this stability makes them attractive candidates for a diagnostic assay.Urine collection is independent of medical facilities, does not require sterile conditions, medical equipment/personnel, and involves minimal sample preparation.Urine is a readily available sample for point-of-care diagnostic tests.


Finally, we have used mass spectrometry (MS) to discover the six leishmanial biomarkers because this technology is proven, powerful and sensitive, and it has been employed for more than 20 years for the detection and precise identification of peptides in complex mixture of proteins [72,73,74]. Therefore, by using MS we warranted a robust and dependable characterization of the leishmanial protein molecules that we have identified in the urine of patients with VL [62,65].

## 3. Approach

Figure 1 illustrates the conceptual idea that urine is a practical biological sample for the discovery of leishmanial biomarkers that can be used for the development of an antigen detection test for the diagnosis of VL. Figure 2 details the general experimental approach that we used for the biomarker discovery and test development.

Using this approach, we firstly identified three parasite biomarkers of VL caused by *L. infantum* and later on three others from *L. donovani* [62,65]. These six markers were used for the development of an antigen detection test for this serious parasitic disease. To investigate and validate the utility of these markers for the diagnosis of VL we initially, as a proof of concept, used a polyclonal antibody-based ELISA. The ELISA was assembled with rabbit IgG antibody (capture reagent) and chicken IgY (developing reagent) specific for each one of the three markers. The strategy to use polyclonal antibodies was fundamentally to validate the antigens as in vivo markers of VL. Polyclonal antibodies are easier to produce, are cheaper and faster to obtain than monoclonal antibodies (mAbs). In addition, we chose to use two animal species to produce the antibodies because in doing so we had a better chance to generate antibodies with a broader range of different antigenic recognition repertoire in these reagents, which is a crucial for assembling a sensitive capture ELISA. Subsequently, to translate these results in the development of an immortalized test for clinical availability we produced and validated a multiplexed mAb-based test.

## 4. Markers Discovery and Polyclonal Antibody-Based Test

### 4.1. L. infantum Markers

The search for *L. infantum* proteins in the urine of VL patients was carried out using samples obtained through our collaborations with Dr. Carlos Henrique Nery da Costa, Instituto de Doenças Tropicais Natan Portella, Federal University of Piaui, Teresina, PI, Brazil and Dr. Ricardo T. Fujiwara, Department of Parasitology, Federal University of Minas Gerais, Belo Horizonte, MG, Brazil. None of the enrolled patients had any clinical signs or symptoms, or laboratory findings compatible with renal or urinary tract abnormalities. These exclusion criteria were important to rule out renal pathology, which could theoretically be a factor that would be biasing the finding of *L. infantum* antigens present in the patients’ urine. None of the patients were under anti-leishmania therapy at the time of urine collection. The mass spectrometry analysis generated a total of approximately 400 peptide sequences. Most of them were of peptides that had identical sequence homologies with those of human proteins. However, eight peptide sequences that had no known homologies with human proteins had identical sequence homologies with the following parasite proteins: *L. infantum* iron superoxide dismutase (*Li-isd1*), NCBI accession XP_001467866.1; *L. infantum* tryparedoxin (*Li- trx1*), NCBI accession XP_001466642.1; and *L. infantum* nuclear transport factor 2 (*Li-ntf2*), NCBI accession XP_001463738.1). The genes coding for these proteins were cloned and the recombinant molecules were produced in *Escherichia coli*. Antibodies to the purified proteins were produced in rabbits and chickens and were used to develop a capture ELISA designed to detect the native *L. infantum* antigens in the urine of VL patients. The assembled capture ELISAs had a sensitivity of 5–45pg/mL of protein (measured with the recombinant proteins). Importantly, human urine did not interfere with the assay sensitivity. We next investigated the possible utility of this ELISA as an antigen detection test for the diagnosis of active VL. Urine samples from 20 VL patients from Brazil were tested and we detected the *L. infantum* antigens in urine samples of 18 of them [45,46,47]. In addition, the assay showed excellent specificity (100%) when tested with urine samples not only from healthy control subjects but from non-VL patients who have other infectious diseases (cutaneous leishmaniasis, Chagas disease, schistosomiasis, and tuberculosis).

### 4.2. L. donovani Markers

Unexpectedly, we observed that the three biomarkers described for *L. infantum* were less abundantly present in the urine of patients with Old World VL. Using a multiplexed capture ELISA assembled to detect those three biomarkers (*Li-isd1, Li- trx1* and *Li-ntf2*) at the same time, 12 out of 25 urine samples from the Old World were positive. These results were somehow unexpected because the discovered *L. infantum* proteins have practically 100% sequence identity with those produced by *L. donovani* [75]. One possible cause for the observed lower sensitivity results could have been related to individual variations of the urine pH. To test this possibility, we measured the pH of all urines and found that none of them had a pH lower than 5.5 or higher than 8.0. In addition, we adjusted with PBS 10X the pH of all the urines to pH 7 and repeated the assay. This procedure did not change the results of the assay with the samples from Old World VL.

Alternatively, these unexpected results could be explained on the grounds of the different serological, pathological, and clinical manifestations that exist between Old World VL and New World VL. For example, conventional serological diagnostic tests to diagnose New World VL using *L. infantum* antigens such as K39 are less sensitive for diagnosing Old World VL caused by *L. donovani*. In addition, *L. donovani* causes disease that is primarily anthroponotic in human adults. Dogs are rarely infected. In contrast, *L. infantum* is a zoonotic pathogen. Dogs and canids in general constitute a major reservoir of New World VL. In humans, the disease affects both adults and children. Moreover, a substantial number of patients who recover from Old World VL treatment will develop a dermatosis commonly known as post-kala-azar dermal leishmaniasis or PKDL. It usually occurs in the Indian Subcontinent and East Africa. As many as 10–20% of Indian cases and 50–60% of Sudanese cases develop PKDL after successful treatment of kala-azar. In contrast, PKDL is extremely rare in patients with New World VL. Consequently, we hypothesized that the leishmanial protein biomarkers eliminated in the urine of patients with Old World VL may also differ from the biomarkers found in patients with New World VL.

To overcome the lower sensitivity of the test to detect Old World VL we procured newer *L. donovani* markers in the urine of patients with VL caused by this parasite and used them to develop antibodies to assemble an ELISA that could add a broader sensitivity to the assay assembled with antibodies against the markers discovered in patients with VL caused by *L. infantum*.

The urine samples from VL patients used for this search were from India and Kenya and were obtained through our collaboration with Dr. Shyam Sundar, (Banaras Hindu University, Varanasi, India) and Dr. Fabiana P. Alves (Drugs for Neglected Diseases initiative, Geneva, Switzerland and Kenya Medical Research Institute, Nairobi, Kenya). For the antigen discovery we used the same protocol that we have previously employed for the identification of the leishmanial proteins in the urine of VL patients from Brazil [45,46]. Nine urine samples from *L. donovani* infected patients from India, and six from patients from Kenya were analyzed. Using this approach, we unambiguously identified and selected three new *L. donovani* proteins for assay development [62]. Specifically, these parasite proteins were: *L. donovani* MaoC family dehydratase (*Ld-mao1*), NCBI accession XP_003858460.1; *L. donovani* Peptidyl-prolyl cis-trans isomerase (*Ld-ppi1*), NCBI accession XP_003858557.1; and *L. donovani* Malate dehydrogenase (*Ld-mad1)*, NCBI accession XP_003864180.1. In addition to these three new proteins, we also found in the urine of these patients the orthologs of the *L. infantum Li-isd1, Li-txn1* and *Li-ntf2* proteins, which are the markers that we originally identified in the urine of the VL patients from Brazil. These latter results confirm our previous observation that indicated that a multiplexed ELISA formatted with antibodies that were specific for *Li-isd1, Li-txn1* and *Li-ntf2* proteins detected these markers in the urine of approximately 50% of patients with Old World VL.

The genes coding for *Ld-mao1*, *Ld-ppi1, and Ld-mad1* were cloned and the recombinant molecules were produced in *E. coli*. Antibodies to *Ld-mao1*, *Ld-ppi1* were produced in rabbits and chickens and were used to develop a capture ELISA. Unfortunately, we could not generate high affinity and specific polyclonal antibodies in either rabbits or chicken against the antigen *Ld-mad1*. However, the ELISA assembled to detect *Ld-mao1* and *Ld-ppi1* was tested using the urine of Old World VL patients (*n* = 45) and healthy control subjects (*n* = 24). The results showed that the sensitivities for the assays for *Ld-mao1* and *Ld-ppi1* were 44.4% and 28.8% respectively. Importantly, several urine samples that were positive for *Ld-mao1* were negative for *Ld-ppi1*. The reverse was also noted, and the complementation of these results yielded a combined sensitivity of 53%. These results highlight and confirm our previous observation that a highly sensitive assay would need to include reagents that detect several different biomarkers preferentially assembled in a multiplexed format.

Therefore, a multiplexed ELISA designed to detect not only *Ld-mao1* and *Ld-ppi1* but also the previously discovered biomarkers *Li-isd1, Li-txn1, and Li-ntf2* was then optimized. A pool of purified rabbit antibodies specific for each antigen was used to coat the plates and a second pool of biotinylated chicken IgY antibodies was used as the detecting reagent.

The multiplexed ELISA was tested using 45 urine samples from Old World VL, 25 healthy controls also from the Old World, and with several control samples from non-VL patients who had other infectious diseases (cutaneous leishmaniasis, *n* = 6; Chagas disease, *n* = 6; schistosomiasis, *n* = 6; and tuberculosis, *n* = 12). The results indicated that the multiplexed assay formatted to detect the five markers greatly increased the sensitivity of the diagnostic test of Old World VL to 82% (Figure 3). Also important was the demonstration that the assay had a specificity of 100%, as no positive result was observed with urine from healthy control subjects or from patients having several other infectious diseases (not shown). We believed that the sensitivity of the assembled multiplexed assay could be potentially higher if we had included the marker *Ld-mad1.* Unfortunately, because the rabbit and chicken anti-*Ld-mad1* antisera that we obtained had low sensitivity/specificity we were unable to assemble an acceptable ELISA for the detection of this antigen.

## 5. Monoclonal Antibody-Based Test

These initial experiments were crucial to define that a specific and sensitive protein-based antigen detection assay for the diagnosis of VL was feasible for development. Although polyclonal antibody-based tests (in contrast to mAbs) are not ideal for production and supply of a clinical test, these reagents were critical to validate the diagnostic utility of the discovered leishmanial protein biomarkers.

Indeed, these preliminary successful proof-of-concept observations were critical for their translation into a de facto antigen detection test for VL that is reproducible and available for clinical use, i.e., a mAb-based assay. mAbs are immortalized reagents, which in contrast to polyclonal antibodies, are suitable for mass scaling-up production and uninterrupted supply for assembling and manufacturing the final test. We initially opted for two different approaches for the generation of these reagents: 1. Recombinant single domain antibodies or VHH derived from immunized camelids; and 2. Conventional hybridoma derived mAbs generated from immunized in mice.

### 5.1. Recombinant Single Domain Antibodies or VHH

VHH antibodies offer a new approach to generating renewable immortalized and monoclonal reagents for diagnostics development. With a molecular weight of approximately 13 kDa, these high affinity, single domain variable fragments of heavy chain antibodies have the ability to refold and retain binding activity after denaturation [76]. VHHs do not have the Fc portion of conventional immunoglobulins, which theoretically are reagents that should provide less background noise in tests like ELISA due to reduced non-specific binding between immunoglobulins (rheumatoid factor-like) [77]. Moreover, large amounts of VHHs can be produced by standard recombinant protein expression in *E. coli*.

As a proof-of-concept we initially developed VHHs to the three *L. infantum* markers *Li-isd1, Li-txn1* and *Li-ntf2*. The immunization of two alpacas with a pool of the three antigens, which resulted in high serum antibody titers to the three antigens was followed by successful screening of a VHH library constructed using B cells obtained from the peripheral blood mononuclear cells from these animals [64,76,78,79,80]. VHH clones specific for *Li-isd1* and *Li-ntf2* were readily detected. Purified soluble recombinant proteins from three different clones specific for *Li-isd1* and from two clones specific for *Li-ntf2* were easily obtained. Unfortunately, and for reasons not clear, only one VHH clone reactive to *Li-txn1* could be retrieved. In addition, this VHH clone showed poor specificity for the antigen. Because several approaches, including capturing the antigen on a plate coated with purified rabbit anti-*Li-txn1* antibody followed by re-screening the library, did not improve the detection of *Li-txn1* VHH clones, we de-prioritize this marker for further proof-of-concept studies.

The capture ELISAs assembled with either anti-*Li-isd1* or anti-*Li-ntf2* VHHs showed similar biochemical sensitivity to that of our former ELISA assembled with rabbit and chicken conventional antibodies (~15–45 pg/mL of urine), therefore compatible with the development of a sensitive clinical diagnostic test for VL. Indeed, the initial clinical validation of the ELISAs assembled with VHHs clearly showed that the selected and engineered reagents had the same clinical sensitivity and specificity for the diagnosis of VL that is similar to that of the assay developed with conventional polyclonal antibodies [64]. The combined sensitivity of the VHH assays to detect *Li-isd1* and *Li-ntf2* was 54.2%, which is only slightly lower than 58.3%, which was the sensitivity of the assay performed with conventional rabbit and chicken Abs.

Although these results confirmed that VHHs can be an interesting alternative for the generation of immortalized reagents for the development of an antigen detection diagnostic test for VL, the fact that we could not generate clones specific for *Li-txn1* was a practical deterrent. Therefore, for this logistic reason we opted, at that time to de-prioritize VHHs and concentrate our efforts on the development of an antigen detection assay assembled with conventional hybridoma derived mAbs.

### 5.2. Conventional mAbs

Subsequent to immunization of mice with the six biomarkers and generation of hybridomas, we performed limiting dilution for their cloning and determination of reactivity of the antibody that they produced with the immunizing antigens. Reactivity was tested by direct ELISA and Western blot. Twenty positive hybridoma clones specific for each marker were selected for mapping of epitope recognition. This step aimed to select clones producing mAbs that would recognize different epitopes in each protein, which is an essential condition for the assembling of a sensitive capture ELISA. The epitope mapping was performed using synthetic 20mer peptides covering the entire full length of each biomarker and overlapping by 10 amino acids. Several mAbs that were specific for different epitopes spanning from the *N* to the *C* terminus of each biomarker were obtained. Only mAbs of the IgG isotype were selected.

For the development of the final capture ELISA, each purified IgG mAb was tested either as a capture or developing reagent using a checkerboard approach, i.e., each of all mAbs specific for one epitope of one of the biomarkers was tested as a capture reagent, paired with all other purified biotinylated mAbs specific for the other epitopes of the same biomarker. Interestingly and surprisingly, the best combination of mAb pairs comprised of antibodies that recognize epitopes that were almost contiguous to each other, in opposition to epitopes that are located far from each other in the protein molecule.

Using this stringent protocol, we were able to assemble highly sensitive ELISA (5–45 pg/mL) for all six biomarkers [61]. When tested with urine samples from VL from the New and Old World, each individual assay, with the exception of the assay assembled to detect *Ld-mad1*, had a sensitivity that varied from 20% to 45%. Many of the urine samples were positive for several markers but some of them were positive for only a single marker. In contrast, *Ld-mad1* was only detected in the urine from VL patients from the Old World. We do not have at this time an explanation for this unexpected result. Nonetheless, the multiple and different recognition of the individual assays confirmed our previous observation that a highly sensitive assay would need to include reagents that detect several different biomarkers at the same time, preferentially assembled in a multiplexed format. In addition, this test format is faster to be performed compared to many single assays and it has a much lower cost for manufacturing.

The multiplexed ELISA designed to simultaneously detect *Li-isd1, Li-txn1, Li-ntf2, Ld-mao1, Ld-ppi1* and *Ld-mad1* was then assembled using two different pools of six specific mAbs, one as capture reagent and the second as biotinylated developing pool. Importantly, the multiplex assay preserved the same sensitivity of the individual assays to detect each biomarker. The initial clinical validation of the multiplexed assay was performed with 69 urine samples from VL from New and Old World and with several control samples from healthy subjects and from non-VL patients who had other infectious diseases (cutaneous leishmaniasis; Chagas disease; schistosomiasis; and tuberculosis). The results are illustrated in Figure 4 and show that the multiplexed assay has excellent sensitivity for the diagnosis of VL (93%). Moreover, the multiplexed assay had a specificity of 100% since no positive result was observed with urine samples from healthy control subjects or from patients having other infectious diseases. As expected, the multiplexed assay assembled with the pool of mAbs had a much higher sensitivity (93%) for diagnose of VL than the assay assembled with polyclonal antibodies (82%). This increment in sensitivity likely occurred not only because of better purity/affinity of the antibodies used in the mAb-based versus polyclonal Ab-based assay but also because it detected one more marker (*Ld-mad1*) than the polyclonal-based assay.

We are in the process of expanding the clinical validation of this promising mAb-based multiplexed test for the diagnosis of VL to translate it into an actual clinical tool. This validation, which will be performed in real time where VL is endemic includes a much larger sample size and will have patients from Brazil, India, and possibly other endemic countries. In addition, it will include equal numbers of healthy control subjects who live in the endemic areas of VL as well as patients with non-VL infectious diseases like cutaneous leishmaniasis, Chagas’ disease, African trypanosomiasis, malaria, schistosomiasis, helminthic diseases, tuberculosis, hanseniasis, etc.

Because an antigen detection test is by definition dependent on the release of specific molecules by actively multiplying microorganisms, this multiplexed developed test should also be a useful resource for diagnosis of not only active VL but also of other clinical forms and/or the severity of the disease e.g., post kala-azar dermal leishmaniasis and VL/HIV co-infection. Moreover, the developed multiplexed ELISA is a very promising tool for following the efficacy of VL treatment, as antigen abundance decreases concomitantly with the elimination of the parasites. We have preliminary evidence to support this proposed utility [81].

Finally, the multiplexed antigen detection assay has the potential to be a very useful tool for the diagnosis of canine VL. However, this utility will be warranted only if the test can be validated using dog’s serum instead of urine. This restriction is based on the fact that from the practical point of view it is not easy to perform random urine collection in dogs as is in humans. We are currently evaluating the validity of the multiplexed test in serum samples for the diagnosis of both canine and human VL.

## 6. Overall Conclusions

This new multiplexed antigen detection assay will be highly valuable for distinguishing active VL from healthy subjects with prior exposure to *L. infantum* or *L. donovani*, an important condition to identify people who require therapy. This assay will be a substantial improvement over the current invasive gold standard test, which is detection of parasite or nucleic acid in liver, spleen or bone marrow aspirates. The test will also constitute a great improvement over current serum tests, which detect anti-leishmania antibodies but cannot distinguish active infection from prior exposure. The existing antigen detection assay is a latex agglutination test based on parasite polysaccharides. The test requires visual reading of agglutination, which is subjective and complicates the distinction between weakly positive from negative results. In contrast, a protein-based ELISA provides clear distinction between positive and negative results.

We are aware that the ultimate goal for development of a more powerful VL test would be a point-of-care (POC) kit that does not require laboratory training or facilities. Newer and sensitive POC technologies are in advanced phases of development and we are in the process evaluating them formulated with the mAbs that we have produced for the development of an accurate rapid test for diagnose of active VL.

## Figures and Tables

**Figure 1 microorganisms-08-01676-f001:**
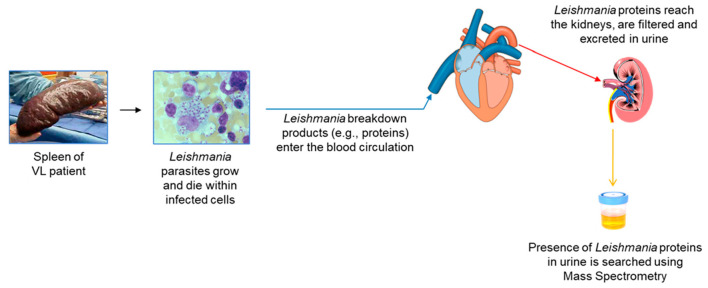
Illustration of conceptual idea and steps used to identify *L. infantum L. donovani* proteins in urine of patients with visceral leishmaniasis.

**Figure 2 microorganisms-08-01676-f002:**
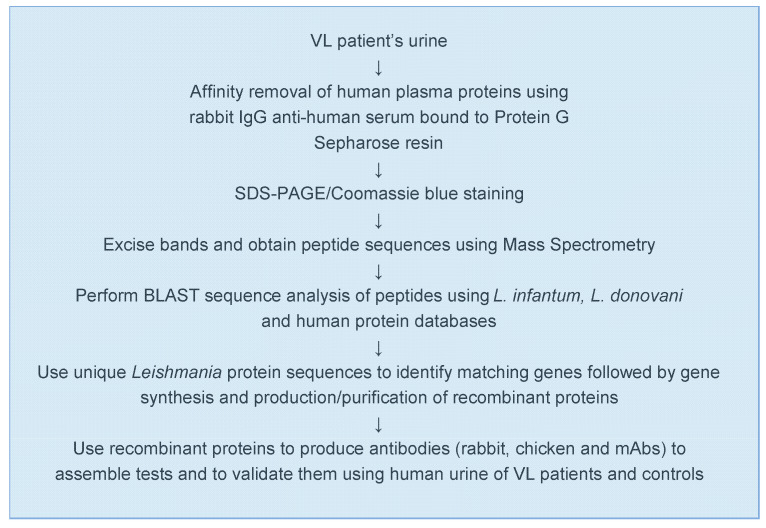
Experimental approach used for the discovery of *L. infantum/L. donovani* biomarkers and test development.

**Figure 3 microorganisms-08-01676-f003:**
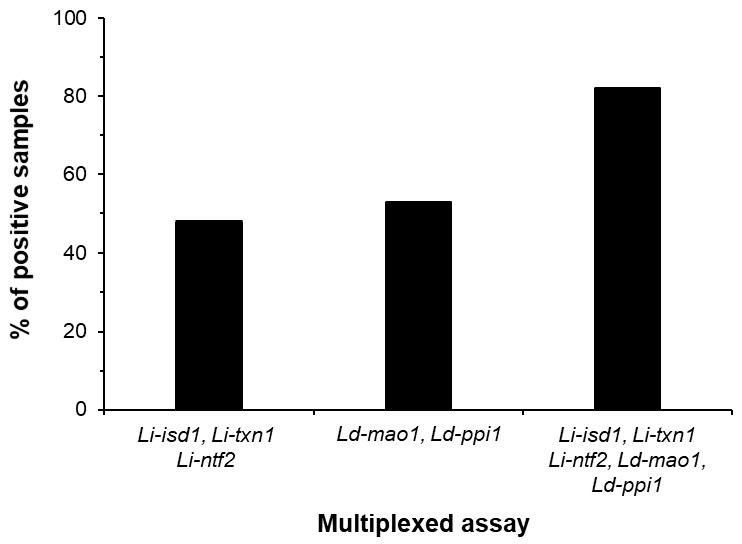
Increment of multiplexed assay sensitivity for diagnose of Old World VL by adding marker detection. ELISA plates wells were separately coated with a pool of affinity purified rabbit IgG antibodies specific for: *Li-isd1, Li-txn1, Li-ntf2* (initially discovered in urine of New World VL patients); *Ld-mao1* and *Ld-ppi1* (discovered in urine of Old World VL patients); and a mixture of antibodies specific for all five markers (*Li-isd1, Li-txn1, Li-ntf2, Ld-mao1* and *Ld-ppi1*). Plates were blocked followed by overnight incubation with urine samples from Old Word VL patients (*n* = 45). Plates were washed and wells were incubated with a second pool containing biotinylated chicken IgY antibodies specific for the leishmanial antigens. Wells were then incubated with streptavidin labeled peroxidase, the substrate H_2_O_2_ and the chromophore TMB. OD was then read at 450 nm. Note the clear assay sensitivity increment with addition of marker detection. The marker *Ld-mad1* was not included in this experiment because no specific polyclonal antibodies to this antigen was successfully generated either in rabbit or chicken. VL, visceral leishmaniasis.

**Figure 4 microorganisms-08-01676-f004:**
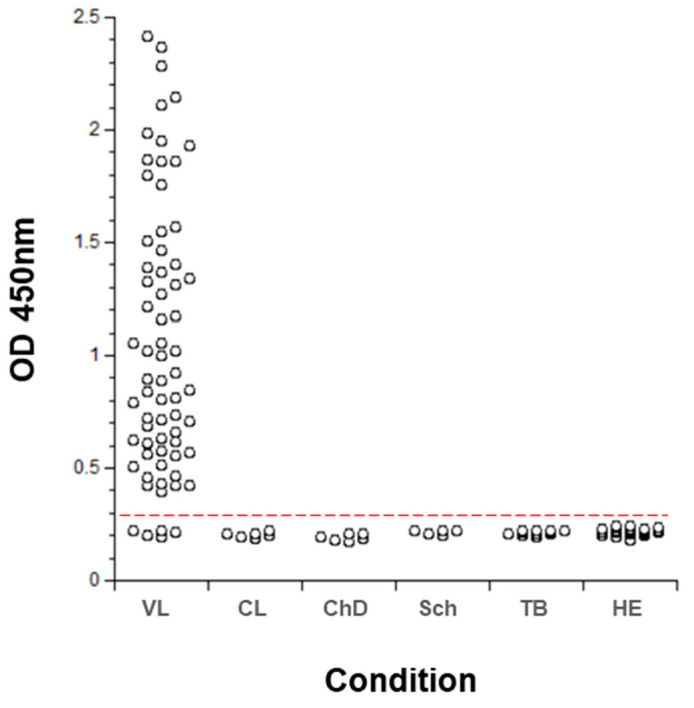
Validation of a new multiplexed assay for the diagnosis of VL. ELISA plates were coated with a pool of mAbs (IgG) specific for all six biomarkers (*Li-isd1, Li-txn1, Li-ntf2, Ld-mao1, Ld-ppi1* and *Ld-mad1*) followed by blocking and overnight incubation with urine samples from VL patients (*n* = 69). Patients were from Brazil (*n* = 24) and Kenya (*n* = 45). Control samples included and urine from non-VL patients with the following diseases: CL (cutaneous leishmaniasis), *n* = 6; CD (Chagas disease), *n* = 6; Sch (schistosomiasis), *n* = 6; and TB (tuberculosis), *n* = 12. In addition, urine from 35 healthy control subjects (HE) were also included. Plates were washed and wells were incubated with a second pool containing biotinylated mAbs specific for the six leishmanial antigens. Wells were then incubated with streptavidin labeled peroxidase, the substrate H_2_O_2_ and the chromophore 3,3′,5,5′-tetramethylbenzidine (TMB). Optical Density (OD) was then read at 450 nm. The dashed red line represents the cutoff value (0.257) which was calculated using the average of the OD obtained from the urine of normal healthy control subjects + 3 SD. No difference in the assay sensitivity (93%) between urine samples from VL patients from New World and Old Word VL was observed when the assay was performed separately with these samples (not shown).

**Table 1 microorganisms-08-01676-t001:** *Leishmania infantum/donovani* proteins found in urine of patients with viscera leishmaniasis.

Protein	Abbreviature	MW (kDa)	NCBI Accession
Iron superoxide dismutase	*Li-isd1*	21.53	*XP_001467866.1*
Tryparedoxin	*Li-txn1*	16.7	*XP_001466642.1*
Nuclear transport factor 2	*Li-ntf2*	13.89	*XP_001463738.1*
MaoC family dehydratase	*Ld-mao1*	16.97	*XP_003858460.1*
Peptidyl-prolyl cis-trans isomerase	*Ld-ppi1*	12.62	*XP_003858557.1*
Malate dehydrogenase	*Ld-mad1*	33.28	*XP_003864180.1*

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
