# Peer review of "Urine-Based Antigen (Protein) Detection Test for the Diagnosis of Visceral Leishmaniasis"

_microorganisms, 2020, doi:10.3390/microorganisms8111676_

Round 1

Reviewer 1 Report

In this review, the authors have summarized the research carried out to identify VL marker proteins in urine and the advances towards the development of a reliable diagnostic system for VL in human patients. The review is comprehensive and well organized. The text is very well written and the bibliographic citations up-to-date and well chosen.

Here are some minor comments for the authors to consider in a revised version of their manuscript.

The abstract is unnecessary long. Please summarize the main matters of the review in a shorter form (author instructions  recommend 200 words maximum).

Regarding the use of the term antigen for proteins isolated from the urine.  In the title and many other parts of the work, the term antigen is used to refer to markers (proteins) isolated from urine. This term generates confusion, since there is no evidence in the text of the review that the host immune system generates antibodies against them. Certainly, they are antigenic in the models used to generate nanobodies or monoclonal antibodies (inoculated in their recombinant forms or synthetic peptides) but not necessarily in patients. My recommendation would be to edit its use throughout the manuscript or to explain why the term antigen is employed to refer to them (Title, abstract; paragraphs 5 and 6 of the first page of the introduction, etc.).

Paragraph 4 page 3 (change details by detail).

Consider editing  the text of the last part of overall rationale section (the text included in page  4). As it is now it seems a draft with some ideas to be included in a definitive text.

Page 6. 2nd paragraph. Homology is an evolutionary relationship of two genes or proteins that come from the same gene (protein), therefore it is not appropriate to use percentages to refer to homology. In this case it would be 100% of sequence identity. Please correct it.

The VHH methodology is very specific. Therefore the authors should consider including a better description of the library construction process, the type of library generated, how it was screened, etc. or alternatively, cite some work that clarifies this methodology.

What about diagnosis of canine VL Do the authors have the intention of evaluating these tools in this pathology that until now lacks a gold standard in diagnosis?

Author Response

Reviewer 1

Comments and Suggestions for Authors

In this review, the authors have summarized the research carried out to identify VL marker proteins in urine and the advances towards the development of a reliable diagnostic system for VL in human patients. The review is comprehensive and well organized. The text is very well written and the bibliographic citations up-to-date and well chosen.

Here are some minor comments for the authors to consider in a revised version of their manuscript.

The abstract is unnecessary long. Please summarize the main matters of the review in a shorter form (author instructions  recommend 200 words maximum).

Answer: The Abstract has been re-written to have the recommended 200 words maximum.

Regarding the use of the term antigen for proteins isolated from the urine.  In the title and many other parts of the work, the term antigen is used to refer to markers (proteins) isolated from urine. This term generates confusion, since there is no evidence in the text of the review that the host immune system generates antibodies against them. Certainly, they are antigenic in the models used to generate nanobodies or monoclonal antibodies (inoculated in their recombinant forms or synthetic peptides) but not necessarily in patients. My recommendation would be to edit its use throughout the manuscript or to explain why the term antigen is employed to refer to them (Title, abstract; paragraphs 5 and 6 of the first page of the introduction, etc.).

Answer:  The suggestion has been accepted and included in the manuscript (page 5, lines 86-88).

Paragraph 4 page 3 (change details by detail).

Answer:  Page 6, line 124.

Consider editing  the text of the last part of overall rationale section (the text included in page  4). As it is now it seems a draft with some ideas to be included in a definitive text.

Answer:  The text has been modified (page 8, lines 168-173).

Page 6. 2nd paragraph. Homology is an evolutionary relationship of two genes or proteins that come from the same gene (protein), therefore it is not appropriate to use percentages to refer to homology. In this case it would be 100% of sequence identity. Please correct it.

Answer:  Correction has been made (page 11, lines 233-234).

The VHH methodology is very specific. Therefore the authors should consider including a better description of the library construction process, the type of library generated, how it was screened, etc. or alternatively, cite some work that clarifies this methodology.

Answer:  We regret that references to the methodology was not included in the submitted manuscript.  References have now been added (page 15, line 336).

What about diagnosis of canine VL Do the authors have the intention of evaluating these tools in this pathology that until now lacks a gold standard in diagnosis?

Answer:  We appreciate the important suggestion.  We are indeed working on the validation of the test utility for the diagnosis of canine VL.  We have included this information in the manuscript (pages 19-20, lines 428-433).

Reviewer 2 Report

The manuscript “Urine-Based Antigen (Protein) Detection Test for the Diagnosis of Visceral Leishmaniasis “ gives very interesting information to find useful and easy teats for the diagnosis of VL. This manuscript could be accepted after minor revision.

  • The introduction is too long and the first part could be deleted because has no interest for this manuscript. Probably the introduction needs to start with “A major impediment to achieve an effective management of visceral leishmaniasis (VL)……” 
  • Use Leishmania in capital and cursive in all the text
  • The authors need to clarify and show, especially in the Discussion and Conclusions, that this promising diagnostic test is pending of a clear clinical validation trough a study with a nice N of samples from active visceral leishmaniasis patients and from both control patients, infected and no.
  • The authors need to reduce and rewrote the conclusions

Author Response

Reviewer 2

Comments and Suggestions for Authors

The manuscript “Urine-Based Antigen (Protein) Detection Test for the Diagnosis of Visceral Leishmaniasis “ gives very interesting information to find useful and easy teats for the diagnosis of VL. This manuscript could be accepted after minor revision.

The introduction is too long and the first part could be deleted because has no interest for this manuscript. Probably the introduction needs to start with “A major impediment to achieve an effective management of visceral leishmaniasis (VL)……”

Answer:  We respectfully disagree with the reviewer.  We have deliberately expanded the Introduction to include a more descriptive background because the journal Microorganisms targets a broad spectrum of scientific areas, which in our view  requires that published manuscripts describe more basic and general information about the microorganisms/diseases/environment that are being studied than that used in journals that have more defined target readers.  Therefore, we would request the Reviewer/Editor to keep the length of the Introduction as is.

Use Leishmania in capital and cursive in all the text

Answer:  Done.

The authors need to clarify and show, especially in the Discussion and Conclusions, that this promising diagnostic test is pending of a clear clinical validation trough a study with a nice N of samples from active visceral leishmaniasis patients and from both control patients, infected and no.

Answer:  This point has been included in the Discussion (page 19, lines 412-419).

The authors need to reduce and rewrote the conclusions

Answer:  Conclusions section has been reduced.